# A New Approach to Optimize SVM for Insulator State Identification Based on Improved PSO Algorithm

**DOI:** 10.3390/s23010272

**Published:** 2022-12-27

**Authors:** Lepeng Song, Qin Liang, Hui Chen, Hao Hu, Yu Luo, Yanling Luo

**Affiliations:** 1School of Electrical Engineering, Chongqing University of Science and Technology, Chongqing 401331, China; 2The School of Electronic Information and Electrical Engineering, Shanghai Jiao Tong University, Shanghai 200240, China

**Keywords:** insulator, state identification, particle swarm optimization, support vector machine

## Abstract

The failure of insulators may seriously threaten the safe operation of the power system, where the state detection of high-voltage insulators is a must for the normal and safe operation of the power system. Based on the data of insulators in aerial images, this work explored an enhanced particle swarm algorithm to optimize the parameters of the support vector machine. A support vector machine model was therefore established for the identification of the normal and defective states of the insulators. This methodology works with the structure minimization principle of SVM and the characteristics of particle swarm fast optimization. First, the aerial insulator image was segmented as a target by way of the seed region growth based on double-layer cascade morphological improvements, and then, HOG features plus GLCM features were extracted as sample data. Finally, an ameliorated PSO-SVM classifier was designed to realize insulator state identification. Comparisons were made between PSO-SVM and conventional machine learning algorithms, SVM and Random Forest, and an optimization algorithm, Gray Wolf Optimization Support Vector Machine (GWO-SVM), and advanced neural network CNN. The experimental results showed that the performance of the algorithm proposed in this paper touched the top level, where the recognition accuracy rate was 92.11%, the precision rate 90%, the recall rate 94.74%, and the F1-score 92.31%.

## 1. Introduction

In the continuous expansion of power grid construction scale, there is a need to take more care of the safety and reliability of transmission lines, which is directly associated with the stability of power transmission. As one of the important components in transmission lines, insulators are of various types and quite often work in the wild for a long period, being vulnerable to environmental pollution, lightning strikes and flashovers. Situations such as string drop and flashover indicate that deteriorated insulators may be found from time to time. Failure of the insulators would cause the power system to fail to operate normally, and even cause grid paralysis in severe cases, bringing great hidden dangers to the safe power supply of the transmission line.

In the past few decades, the detection of insulator faults relies on manual inspection, at regular intervals or yearly inspection on deteriorating insulators. Although faulty insulators can be detected and replaced in time, the work intensity is high, and the efficiency is low. This is complicated with the detection location at high altitude, which somehow affects the personal safety of the detection personnel, as shown in Figure 1a.

In recent years, IT technology has been widely popularized, making it possible to have unmanned aerial vehicle (UAV) [1,2,3,4] as a core means of power transmission line inspection. This is a kind of inspection methodology highlighted as accurate, safe and efficient, as shown in Figure 1b. The inspectors may shoot and store a large amount of image and video data through the camera mounted on the drone, including digital image information, infrared image information, etc. Owing to this, the application of digital image technology has contributed a lot in the power safety system. These pictures processed by image technology may perform state detection on the basis of automatic positioning of insulators and diagnose insulator faults in a timely and effective manner.

The state detection of insulators is taken as one of the important links to ensure the normal operation of transmission lines. Due to the special working environment of insulators and the complex background of aerial photographs, accurate identification and segmentation of target insulators and background images can be the primary prerequisite for detecting their states. When segmenting insulators, Ke et al. [5] designed an image segmentation method based on weighted variable FCM, following pixel differences to separate the insulators and transmission lines. Wu et al. [6] Proposed a global minimum frame (GMAC) for insulator segmentation using texture features. Yu et al. [7] set their focus on its shape and texture characteristics; then, the segmentation of insulators was achieved by alternating texture and shape driving curves. Yin et al. [8] worked out a new edge detection operator based on double-parity morphological gradients to segment infrared insulator images, and the average success rates of segmenting insulators with voltage levels of 110 and 220 KV reached 98.3% and 96.4%, respectively. As to the same infrared image, Wang et al. [9] used Mask R-CNN to achieve insulator segmentation. Cui et al. [10] extracted the features of different levels of insulators and added the ED network to make sure that the feature fusion module works better, thereby realizing the segmentation of insulators.

At present, neural networks are extensively applied in the fault identification and classification of insulators. Prates et al. [11] elaborated a convolutional neural network (CNN) to identify defects in insulators, and the defect detection accuracy was as good as 85.48%. Guo et al. [12] started with a deep convolutional neural network to extract features and then turned to an enhanced AlexNet model and SVM to detect and classify transmission line anomalies. Jiang et al. [13] constructed a multi-layer perception architecture for fault detection on insulators missing a cap, obtaining an accuracy rate of 91.23. There are still many researchers that have made efforts in detecting faults of insulators by improving Faster RCNN [14,15,16,17,18]. Han et al. [19] and Liu et al. [20] made different improvements on the yolo network to achieve fault detection of insulators. Yu et al. [21] extracted features that were fused with five CNN networks, followed by combining with RF to detect foreign objects in transmission lines; additionally, semi-protocol deep neural network [22], echo state network [23], Kear model [24], and fully convolutional neural network (u-net) [25,26] have all brought out some impressive outputs in insulator defect detection.

Despite that positive feedback from various studies based on deep learning and long-term training still needs a large amount of data, for small data samples, machine learning is believed to have a better outcome. For instance, in the identification of cable faults, Wang et al. [27] proposed to use an improved particle swarm optimization support vector machine (IPSO-SVM) algorithm with a 91.9% identification accuracy. Pernebayeva et al. [28] classified the surface images of insulators in different environments and compared them with neural networks to select reasonable features. The conventional machine learning algorithm presented higher classification accuracy. Machine learning methods were also very successful in the recognition of handwritten characters. Kundu et al. [29] realized the recognition of handwritten keywords by extracting angle features through Hough transform. Kang et al. [30] proposed an unsupervised redundant co-clustering algorithm (FCMSC) based on multi-center fuzzy c-means clustering (FCM) and spectral clustering (SC) in the detection of mildew distribution of corn kernels, with an accuracy of 93.47%. Corso et al. [31] investigated the application of machine vision features and classified the degree of contamination of insulators based on k-nearest neighbors (k-NN) with an average accuracy of over 82%. Yao et al. [32] explored the classification of insulation defects in gas-insulated switchgear and then operated SVM to identify the classification, gaining its accuracy rate of 93.75%. Sun et al. [33] applied SVM to abnormal conductive faults in conductive copper rods recognition, bringing the accuracy rate up to 90%.

In insulator defect identification research, SVM is one of the most commonly used methods for small-sample insulator datasets. Murthy et al. [34] extracted the features of insulators by means of wavelet transform and then used SVM to identify the state of insulators. Reddy et al. [35] obtained the features of insulators with the help of discrete orthogonal transform (DOST), allowing SVM to identify insulators. Yan et al. [36] first fused HOG and LBP features after PCA dimensionality reduction and then counted on SVM to detect and classify insulators. Sun et al. [37] employed the EFA method to reduce the factor variables of insulator pollution, taking the simplified factor variables as new input variables, and the LSSVM model was established to predict the pollution degree of insulators. Ma et al. [38] introduced the gray level co-occurrence matrix and Tamura features that were applied in the SVM classifier to identify the discharge degree of pollution flashover insulators, getting the classification accuracy as high as 90.6%.

SVM application can be found in extensive research fields. Scholars have made various improvements on the conventional SVM classifier, so that the corresponding research has achieved good results. The SVM research in different fields is shown in Table 1.

As for the defect detection research of insulators, there are few representative and public datasets, and the background of aerial insulator images appears to be very complex. For conventional machine learning, the segmentation target insulator image must be identified first, which is relatively difficult. Third, compared with aerial images, the shedding defects of the insulator strings are too small, and it is difficult to accurately extract and detect the modified features. In this paper, targeting at a few aerial insulator pictures, an enhanced particle swarm algorithm (PSO) is proposed to optimize the SVM to obtain the training model and identify and classify the insulator state. The contributions of this work to the state detection of insulators in high-voltage transmission lines are summarized as follows:

(1) In the segmentation of insulator images, the segmentation of insulators under complex backgrounds was accomplished by introducing a double-layer cascaded morphological structure into the conventional seed region growing algorithm.

(2) These features of HOG and GLCM were extracted and fused each other. The fused features were input into the boosted PSO-SVM training model to successfully develop a novel method for insulator fault identification and classification. This method enforced the generalization performance of SVM and may be applied to the field of high-voltage transmission lines.

(3) As to small data aerial insulator images, the proposed model, e.g., the boosted PSO-SVM model, showed impressive accuracy when compared with deep learning models such as CNN and BP, as well as other conventional machine learning algorithms such as SVM and Random Forest.

The structure of the rest of this paper is organized as follows: Section 2 presents the overall architecture of insulator state detection. Section 3 describes the specific process of insulator fault diagnosis. Section 4 reveals how experiments and results analysis were performed. Finally, Section 5 gives the conclusion.

## 2. Research Methods

In this section, the framework and related algorithm theory for the state detection of transmission line insulators are presented. The overall framework of this paper is shown in Figure 2. Due to the complex background of aerial insulator images, the first step is to separate the insulator image and the background image. As proposed in this paper, the seed region growing algorithm is improved to effectively segment the insulator, then extract the features of the insulator image, use the feature data to train the particle swarm parameters to optimize the support vector (PSO-SVM) classifier, and finally the trained PSO-SVM classifier may accurately complete the state detection of insulators.

### 2.1. Region Algorithm Segmentation Based on Morphological Improvements

The aerial insulator images often come with a complex background. To ensure the effectiveness of feature extraction, it is necessary to separate the background and retain only the insulator region. The insulator image had obvious regional characteristics, and the seed region growing algorithm was therefore selected to realize the segmentation of the insulator image. Note that the direct use of this algorithm may retain the large or small background that is similar to the gray value of the insulator region. To tackle this issue, this work adopted a novel algorithm for seed region segmentation based on morphological improvements, and its overall block diagram is shown in Figure 3.

The insulator segmentation algorithm proposed in this work firstly reduced the dimension of the original image, selected the appropriate color component model, then performed morphological processing on it, and finally used the seed region growth algorithm to segment to achieve effective segmentation of the insulator region. The specific process is shown in Figure 4.

The aerial insulator images were colored and stored in RGB. Since most of the insulators in this work were red relative to the background and displayed high saturation, the color in the HSI space was more conducive to image recognition than that in this work. Then, the insulator image was converted from the RGB model to HSI color model for processing purposes. By observing the HSI component image, it was found that the insulator area in the S component image was prominent. Subsequently, the S component of the insulator was extracted for image segmentation to ensure the segmentation effect while reducing the image dimension and improving the operation speed.

Morphological processing is a series of image processing operations based on shape, which produces an output image by applying structural elements to the input image. The most basic morphological processing is the erosion and dilation operations, and the expressions are as follows:(1)XΘS={x|S[x]⊆X}
(2)X⊕S={x|S[x]∩X≠Φ}

Here, X is the target image; S is the structural element; x indicates the current position. The corrosion operation is mainly used to extract the backbone information of the insulator, and the expansion is used to fill in the edge information of the insulator.

This work was processed with a two-layer cascade morphological structure, which cascaded two structural elements with different structures and different sizes to process images. This promoted the segmentation effect of the conventional seed region growing algorithm: first, to choose a 4 × 4 square structure to perform the corrosion operation on the S-component image, and then to perform the expansion operation on the etched image with a 3 × 3 circular structure to obtain the double-layer cascaded morphological structure. Then, the seed region growing method is used to segment the image.

### 2.2. Feature Extraction

In the computer vision-based insulator identification method, the insulator features refer to those including the insulator image. Features are used to distinguish the states of insulators, mainly including Histogram of Oriented Gradients (HOG) and Gray Level Co-occurrence Matrix (GLCM). These are not newly proposed features, which are present in many fields. In this work, advanced features were mainly used in the identification and classification of insulator states, aiming to improve the accuracy of the classifier.

#### 2.2.1. HOG Feature Extraction

HOG is a local region-based feature descriptor. In past studies, HOG always had a place in many recognitions, such as the recognition and classification of ships and vessels [50], human detection [51], face recognition [52], and note recognition [53]. The HOG feature refers to the use of the gradient information of the local area to represent the edge features of the object, which may well describe the edge features of the insulator. How the insulator HOG feature was extracted is shown in Figure 5.

The formula for calculating the gradient size and direction of an image is as follows:(3)$$G_{x}(x,y)=H(x+1,y)-H(x-1,y)$$
(4)$$G_{y}(x,y)=H(x,y+1)-H(x,y-1)$$
(5)$$G(x,y)=\sqrt{G_{x}{\left(x,y\right)}^{2}+G_{y}{\left(x,y\right)}^{2}}$$
(6)$$\alpha (x,y)=\arctan\left[G_{y}(x,y)/G_{x}(x,y)\right]$$
where G(x,y), $G_{x}(x,y)$, $G_{y}(x,y)$, and H(x,y) are the gradient value, horizontal gradient, vertical gradient and pixel value of the current pixel, respectively, and α(x,y) is the gradient direction.

#### 2.2.2. GLCM Feature Extraction

GLCM described the texture features of objects by means of five statistics, including energy, entropy, moment of inertia, correlation, and local stationarity. The grayscale co-occurrence matrix indicated the joint frequency distribution of two grayscale pixels with a distance of (Δx,Δy) in the image, and it can reflect the overall information of the image grayscale in various directions, adjacent intervals, and amplitude changes.

Normalizing the grayscale matrix:(7)$$P(i,j,d,\theta )=\frac{P(i,j)}{\sum_{i=0}\sum_{j=0}p(i,j)}$$

In the formula, P(i,j) indicates how many adjacent paired points there were in the image gray-level of i and j. θ means four directional values of 0°, 45°, 90°, and 135°. The GLCM characteristics of the insulator were obtained by calculating the values of five statistics in four different directions and offsets and by taking their mean and variance.

### 2.3. PSO-SVM

#### 2.3.1. Support Vector Machine Algorithm

Support Vector Machine (SVM) is an algorithm that can intelligently classify data based on supervised learning, providing better solutions to the problems of small samples. The main idea of SVM is to find an optimal hyperplane that maximizes the distance between two different classes of data points, as shown in Figure 6.

In two-dimensional space, a data sample set is $\left(x_{i},y_{i}\right)$, i=1,2,…,l, where the input vector $x_{i}\in R^{n}$, $y_{i}\in [1,-1]$ is the output category and is the number of samples. When the data are linearly separable, there are: $y_{i}(w\cdot x_{i}+b)\geq 1\;i=1,2,\ldots ,l$; now, the optimization problem of SVM is transformed into:(8)$$\{\begin{array}{c} \min\frac{1}{2}{\left\|w\right\|}^{2}\\ s.t.\;y_{i}(w\cdot x_{i}+b)\geq 1 \end{array}$$
where w is the weight vector of the hyperplane; b is the bias term.

When the data samples are linearly inseparable, slack variables $\varepsilon_{i}\geq 0$ and penalty parameters C need to be introduced on the basis of the above formula to construct the optimal classification plane. Thus, the optimization problem is transformed into:(9)$$\{\begin{array}{c} \min\frac{1}{2}{\left\|w\right\|}^{2}+C\sum_{i=1}^{l}\varepsilon_{i}\\ s.t.\;y_{i}(w\cdot x_{i}+b)\geq 1-\varepsilon_{i} \end{array}$$

To solve the optimization problem, the Lagrangian function may be introduced and then transformed into the corresponding dual problem. The Laplace function is expressed as follows:(10)$$L(w,b,\varepsilon_{i},\alpha ,\beta )=\frac{1}{2}{\left\|w\right\|}^{2}+C\sum_{i=1}^{l}\varepsilon_{i}+\sum_{i=1}^{l}\alpha_{i}[1-\varepsilon_{i}-y_{i}(wx_{i}+b)]-\sum_{i=1}^{l}\beta_{i}\varepsilon_{i}$$

In the above formula, $\alpha_{i}$ and $\beta_{i}$ are Lagrange multipliers, and $\alpha_{i}\geq 0,\beta_{i}\geq 0$. The optimization problem is changed to the value of w,b when the Lagrangian function obtains the minimum value; thus, the following formula can be obtained by derivation of $w,b,\varepsilon_{i}$ in the Lagrangian function:(11)$$\{\begin{array}{c} w=\sum_{i=1}^{l}\alpha_{i}y_{i}x_{i}\\ \sum_{i=1}^{l}\alpha_{i}y_{i}=0\\ C=\alpha_{i}+\beta_{i} \end{array}$$

Putting the above formula into the Lagrangian function expression, the following expression can be obtained by simplification:(12)$$L(w,b,\varepsilon_{i},\alpha ,\beta )=-\frac{1}{2}\sum_{i=1}^{l}\sum_{j=1}^{l}\alpha_{i}\alpha_{j}y_{i}y_{j}x_{i}x_{j}+\sum_{i=1}^{l}\alpha_{i}$$

At this point, the Lagrangian function can be converted into a dual problem, that is, to maximize the above formula. The constraints of the dual problem are:(13)$$\{\begin{array}{c} \sum_{i=1}^{l}\alpha_{i}\beta_{i}=0\\ 0\leq \alpha_{i}\leq C \end{array}$$

At this point, according to the combination of the KKT conditions and the existing conditions, we can obtain:(14)$$\{\begin{array}{c} w=\sum_{i=1}^{l}\alpha_{i}y_{i}x_{i}\\ b=y_{i}-x_{j}\cdot \sum_{i=1}^{l}\alpha_{i}y_{i}x_{i} \end{array}$$

The values of w and b are obtained, and the optimal hyperplane decision function is:(15)$$\begin{array}{ll} f(xi) & =\mathrm{sgn}(w\cdot x_{i}+b)\\ & =\mathrm{sgn}(\sum_{i=1}^{l}\alpha_{i}y_{i}(x_{i}\cdot x)+b) \end{array}$$

Support Vector Machines usually solve the case where the data samples are separable, but things are different when faced with the problem of nonlinear data samples. Here, it is necessary to convert the low-dimensional space into a high-dimensional feature space through a nonlinear transformation φ into a linear problem. Of course, in the process of spatial transformation, the problem of data dimensionality disaster requires the introduction of a kernel function. As long as the kernel function satisfies the Mercer condition $k(x_{i},x)=(\varphi (x_{i})\cdot \varphi (x))$, the nonlinear decision function will become:(16)$$f(x_{i})=\mathrm{sgn}(\sum_{i=1}^{l}\alpha_{i}y_{i}K(x_{i},x)+b)$$

#### 2.3.2. Improved PSO Algorithm

As one of the common optimization algorithms of particle swarm, proposed by Kennedy and Eberhart in 1995, it works by simulating the foraging behavior of birds to achieve the group optimization. Reference [54] describes the PSO algorithm in detail. During the optimization process, the velocity and position of the updated particles are as follows:(17)$$v=w\cdot v+c_{1}r_{1}(p_{best}-p)+c_{2}r_{2}(g_{best}-p)$$
(18)p=p+β⋅v
where $g_{best}$: global optimal solution; $p_{best}$: individual optimal solution; p is the current position of the particle; v is the velocity of the particle; β is the constraint factor; $c_{1}$ and $c_{2}$ are two positive numbers, respectively, called acceleration factors; $r_{1}$ and $r_{2}$ are two independent random numbers between values [0, 1]; w is the inertia weight.

In practical applications, PSOs tend to fall into local extreme points. To deal with such problems, the inertia weight was improved by absorbing the control idea of adaptive adjustment. Meanwhile, the current number of iterations and the number of populations during the algorithm update iteration were also taken into account with the formula of inertia weight w as follows:(19)$$w=(w_{\max}-w_{\min})\times \frac{\left(\max gen\times sizepop-q\times p\right)}{\max gen\times sizepop}+w_{\min}$$
where $w_{\max}$ represents the maximum inertia weight value, $w_{\min}$ represents the minimum inertia weight value. maxgen indicates the maximum number of iterations, sizepop is the population number, and q, p respectively represent the current iteration and the current population.

#### 2.3.3. Building the PSO-SVM Classifier

Common kernel functions of SVM include: linear kernel function, RBF (radial basis) kernel function, polynomial kernel function and multi-layer perceptual kernel function. In this work, the RBF kernel function was selected as the inner product kernel function. Since the RBF kernel function provides great flexibility for nonlinear mapping of input data, it is useful for complex, nonlinear and inseparable classification problems. It is expressed as:(20)$$K(x,x_{k})=\left\{-\frac{{\left\|x-x_{k}\right\|}^{2}}{2\sigma^{2}}\right\}$$
where $\left\|x-x_{k}\right\|=\sqrt{\sum_{k=1}^{n}{\left(x^{k}-x_{i}^{k}\right)}^{2}}$ and σ are the core widths.

Penalty factor *C* and kernel parameter σ are vital parameters that may affect the accuracy of SVM. In this work, the improved PSO algorithm was used to select the SVM parameters, and thus, the PSO-SVM classifier was obtained, along with the improved PSO optimization SVM parameter process as follows:

Step 1: Initialization settings, including group size, number of iterations, and randomly given *C* and σ as the initial positions of particles;

Step 2: Using the *C* and σ corresponding to the individual particle, the SVM classifier to predict the output value of the test sample was taken as the fitness value of the individual particle $y_{i}$;

Step 3: Compare the optimal adaptive values of the particles $y_{i}$ and $y_{pbesti}$ themselves. If $y_{i}<y_{pbesti}$, replace, respectively, the previous adaptive value and particle with the new adaptive value, namely $y_{pbesti}=y_{i}$, $x_{pbesti}=x_{i}$;

Step 4: Compare the optimal adaptive values of all particles $y_{pbesti}$ and $y_{gbesti}$. If $y_{pbesti}<y_{gbest}$, replace $y_{gbest}$ with $y_{pbesti}$, and keep the current state;

Step 5: Determine whether the adaptive value or the number of iterations meets the required value. If not, update the state through Equations (17) and (18), and then, return to Step 3, or complete the calculation to find the most suitable parameters for SVM *C* and σ.

## 3. State Recognition Method of Insulator Based on PSO-SVM

The insulator state identification method based on the improved PSO-SVM proposed in this paper is shown in Figure 7. The specific implementation is as follows:

Step 1: Preprocess the aerial insulator image to obtain the segmented insulator image;

Step 2: Extract the HOG and GLCM features of the insulator to obtain the sample feature set data, followed by correcting and preprocessing the sample data, and divide the sample data into two parts: training samples and test samples;

Step 3: Use the training samples to train the PSO-SVM classifier to obtain the best PSO-SVM classifier;

Step 4: The test sample was served as the input data of the optimal PSO-SVM classifier, and the insulator state was detected. The completed PSO-SVM is used to evaluate the insulator classification state.

**Figure 7 sensors-23-00272-f007:**
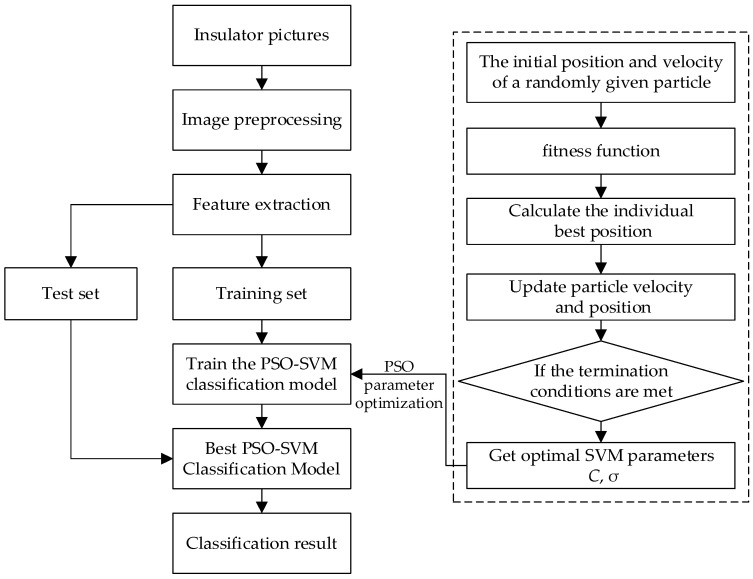
Overall flow chart of insulator status identification.

## 4. Experiment and Result Analysis

### 4.1. Dataset and Experimental Environment

In this work, insulator images were chosen for simulation experiments. The insulator dataset was obtained partly from the GitHub website and partly by post-processing in the field, and the insulator dataset mainly contained images of insulators in normal and damaged states, as shown in Figure 8. This experiment mainly selected 212 random images from the website for the training model and then selected 76 random images taken in the field for the model test; the dataset samples are shown in Table 2.

The experimental software platform was MATLAB, mainly based on the LIBSVM software package developed by Professor Lin Zhiren of National Taiwan University for programming experiments. The specific experimental environment configuration is shown in Table 3.

### 4.2. Image Segmentation

The result of segmentation of the insulator example image is shown in Figure 9. Here, the image segmentation algorithm proposed in this paper was tested, that is, the region segmentation method based on morphological improvements was compared with the conventional seed region growing method and the maximum inter-class variance threshold segmentation method.

Figure 9 shows the segmentation results of normal insulator images and defective insulator images under different algorithms. Image (1) in (a) and (b) is the original image; (2) is the image segmented by means of the maximum inter-class variance threshold. Image (3) is the image segmented by the conventional seed region growing algorithm; (4) is the image segmented by the region segmentation method based on morphological improvements as proposed in this paper. It is clear from the two sets of images in Figure 4 that the segmentation method given in this work was able to effectively segment the insulator image and the background, while the other two methods failed to effectively identify the target image.

### 4.3. Feature Fusion Results and Analysis

To verify the effectiveness of the proposed fusion of HOG and GLCM features in insulator state recognition and classification, a comparative experiment with a single HOG feature and GLCM feature was conducted. During the experiment, the parameters of the particle swarm algorithm were set as follows: the number of particle swarms was 10; the vector dimension was 2; the maximum number of iterations was set to 200; $c_{1}=1.6$; $c_{2}=1.7$; the search range of the parameter *C* was [0.1, 100]; the search range of σ was [0.1, 100]. The experimental results are shown in Table 4.

In the process of insulator classification, the choice of penalty factor C and kernel function of the σ SVM plays a very important role in the classification accuracy of the model. If the value of parameter C is too large, the data will be “over-fitted” and the generalization performance of the model will be reduced, while if the value of parameter C is too small, the model will be “under-fitted”; for σ, too large a value will result in “underfitting” of the data, and too small a value will result in “overfitting”.

In order to avoid the phenomenon of two important parameters of the SVM being too large or too small, the PSO algorithm performs an intelligent search for the SVM parameters. From the performance comparison between the single feature algorithm and the fused feature algorithm for the identification and classification of insulators in PSO-SVM given in Table 4, it can be seen that the HOG, GLCM and fused features used in the classification model yield different combinations of SVM parameters, and thus, different classification results were obtained. The accuracies of HOG, GLCM and fused features are 78.95%, 50.00% and 92.11%, respectively. It can be concluded from the accuracy of the model classification that the fused features algorithm is more effective and accurate than single features in insulator classification.

From the above experimental results, it can be concluded that the fusion features of the test samples are used in the PSO-SVM insulator classification model with certain accuracy. In total, 76 images were randomly selected from 200 test images at a time during the testing process, but since the data of the whole process are small, it is difficult to judge whether the model has overfitting phenomenon; therefore, the model fitting problem is described here by the learning curves so as to judge whether the model has overfitting or underfitting phenomenon.

From the learning curve in Figure 10, we can see that the accuracy of the training set increases and then decreases as the training set increases, and finally stabilizes at around 94%, while the accuracy on the cross-validation set keeps increasing and finally stabilizes at around 92%. The difference between the classification accuracy of the classifier in the training and validation sets is very small, and the accuracy of the test set in the model is 92.11%; thus, the generalization ability of the PSO-SVM model is good and there is no overfitting phenomenon.

### 4.4. Model Classification Comparison

The comparison of PSO-SVM with other models was mainly involved with three aspects: machine learning, adding optimized models and deep learning. Machine learning methods, such as conventional SVM and Random Forest (RF), were compared with the algorithm PSO-SVM in this work; in optimization, the SVM model (GWO-SVM) was optimized by the gray wolf algorithm and compared with the PSO-SVM model; the PSO-SVM algorithm was also put in contrast with the classic classification models Convolutional Neural Network (CNN) in deep learning.

#### 4.4.1. Model Evaluation

In machine learning, the classification model is mainly evaluated with four indicators: accuracy, precision, also known as sensitivity, recall, also known as specificity, and F1-score.
(21)$$Accuracy=\frac{TP+TN}{TP+FN+FP+TN}$$
(22)$$\mathrm{Precison}=\frac{TP}{TP+FP}$$
(23)Recall=TPTP+FN
(24)$$F1\_score=2\times \frac{P\times R}{P+R}$$
where *TP* is the number of normal samples predicted to be normal; *FN* is the number of normal samples predicted to be dropped; *FP* is the number of dropped samples predicted to be normal; *TN* is the number of dropped samples predicted to be dropped.

#### 4.4.2. Comparison of PSO-SVM and Machine Learning Models

The test set samples were used to validate the model in SVM, Random Forest and the proposed model in this study, respectively. As seen in Table 5 and Figure 11, the classification result of the PSO-SVM model is more accurate than that of SVM and Random Forest, the overall performance of SVM was higher than that of Random Forest, while the accuracy rate of the PSO-SVM model went up to 92.11%, and the precision rate, recall rate and F1-score also increased to 90%, 94.74% and 92.31%, respectively. This is significantly better than the conventional SVM algorithm and RF classification algorithm. The overall classification performance of the model proposed in this work excelled in all.

In order to see the classification results of each model more clearly and intuitively, Figure 12 shows the comparison of the recognition and classification results on three models, SVM, Random Forest and PSO-SVM, for the same image selected randomly from the test set.

As can be seen from Figure 12, (a1), (b1) and (c1) are the results of a normal insulator picture identified in each of the three models, of which the SVM and PSO-SVM models identify and classify correctly; (a2), (b2) and (c2) are a defective insulator picture, and all three models can identify and classify correctly; (a3), (b3) and (c3) are the results of the same defective insulator picture for classification, of which only the PSO-SVM can identify and classify correctly.

#### 4.4.3. Comparison of Optimization Algorithms

This paper focuses on the optimization of SVMs using the improved PSO algorithm. The Gray Wolf algorithm (GWO) is one of the many intelligent optimization algorithms that are used to optimize the SVM model for comparison with the improved PSO-SVM model in this paper. In the PSO search for optimization of SVM parameters, the cross-validation fold k = 5, and the minimum error rate of the model is the objective function; for a more effective comparison, the parameters of the gray wolf algorithm are set in the same way as the PSO parameters, such as a wolf pack size of 10, a maximum number of iterations of 200, a search range of [0.1, 100] for all parameters, a cross-validation fold k = 5, and an objective function: minimum error rate, etc. The model performance results of the gray wolf algorithm and the two algorithms after optimizing the SVM are shown in Table 6.

Table 6 shows the classification results of the test set data on the GWO-SVM model and the PSO-SVM model in terms of accuracy, precision, recall, F1 score and average time. Except for the time cost where the PSO-SVM model performs lower than the GWO-SVM model, the PSO-SVM model outperforms the GWO-SVM model in terms of overall performance analysis. The confusion matrix for the test set classification on the PSO-SVM model and the GWO-SVM model is shown in Table 7.

Figure 13 shows the comparison of the recognition and classification results of the same image on the two models. The comparison of specific images provides a more intuitive and effective demonstration of the effectiveness of the algorithm. Images (a1) and (b1) are images of the same normal insulator and are incorrectly identified and classified by both models; (a2) and (b2) are the identification results of a normal insulator image on both models, where the PSO-SVM model completes the correct identification and classification; (a3) and (b3) are images of defective insulators and are correctly identified and classified by both GWO-SVM and PSO-SVM; (a4) and (b4) are comparisons of the results of the same defective insulator image run on the two models, where only the PSO-SVM model is able to recognize and classify correctly.

#### 4.4.4. Comparison of PSO-SVM and Convolutional Neural Network Model

The PSO-SVM model is used in this paper for state recognition classification of insulators. For comparison in other ways, the PSO-SVM model is compared here with convolutional neural networks (CNN). The comparison is made by accuracy (Ac), sensitivity (SP) and specificity (SE). The network architecture of CNN is as follows: an input layer, mainly used for color image input, with an image size of 256 × 256 × 3. Three convolution blocks are used to extract image features. Each convolution block consists of a convolution layer, a normalization layer, a ReLU layer and a pooling layer. The convolution core size of the three convolution layers is 3 × 3. The first layer has 8 convolution cores with a step size of 1, the second layer uses 64 convolution cores, the third layer uses 128 convolution cores, and each of them has a pool layer with a core size of 2 × 2. The last three layers are full connection layer, SoftMax layer and an output layer for classification. The results of the comparison of the two models are shown in Table 8 and Figure 14.

It is clear from Figure 14 and Table 8 that the performance of CNN in insulator recognition and classification is significantly lower than that of PSO-SVM model. The comparative experiments show that the algorithm proposed in this paper is feasible. One normal and one defective image each were randomly selected from the test set to compare the recognition and classification effects of the two models, which are shown in Figure 15, (a1) and (b1) are normal insulator images, which can be correctly identified and classified only by PSO-SVM; (a2) and (b2) show that the defective insulator images run on CNN and PSO-SVM, respectively, and both models can be identified and correctly classified.

## 5. Conclusions and Next Steps

At present, the identification of insulator shedding state is largely the job of conventional manual detection and a neural network. Manual detection of insulator shedding involves a large workload and high risk. The neural network method usually requires a large number of samples for training. In the case of limited samples or small samples, the neural network is incapable of performing the entire identification work. The support vector machine algorithm (SVM) is able to handle the issues in nonlinear, high-dimensional small sample classification, with the recognition accuracy being much higher than the neural network. The only holdback is the selection of the kernel parameters and penalty factors of the support vector machine. PSO is an optimization algorithm that works with particle swarms to optimize the parameters of support vector machines. Nevertheless, due to the problems of convergence speed and local minimum in the PSO algorithm, there are advanced PSO algorithms that have overcome the problems of the original PSO algorithm. Therefore, this paper proposes an insulator state identification method based on the improved PSO algorithm to optimize the support vector machine. This work is concluded as follows:By collecting insulator pictures, preprocessing, feature extraction and selection, the insulator state recognition dataset was obtained. From the recognition results of the model, it can be concluded that this dataset stood out with excellent training performance.The improved PSO was used to optimize the parameters of SVM, which solved the problem that the PSO tended to fall into a local minimum. By avoiding the inappropriate selection of SVM parameters, the performance of the SVM recognition model was optimized.The results showed that the PSO-SVM insulator state recognition and classification model, as proposed in this paper, were compared with machine learning methods, neural network models and optimization algorithms. The final results proved that the PSO-SVM model was proud of the highest classification performance.

Currently, deep learning is widely used in machine vision research and has achieved very good results. There are many mature network models that can achieve better results in image recognition, such as the classical ResNet network and Yolo network. Therefore, the focus of our next research will shift to known architectures, such as ResNet and Yolo, and will use the existing data to conduct deeper research on top of these traditional classical networks.

## Figures and Tables

**Figure 1 sensors-23-00272-f001:**
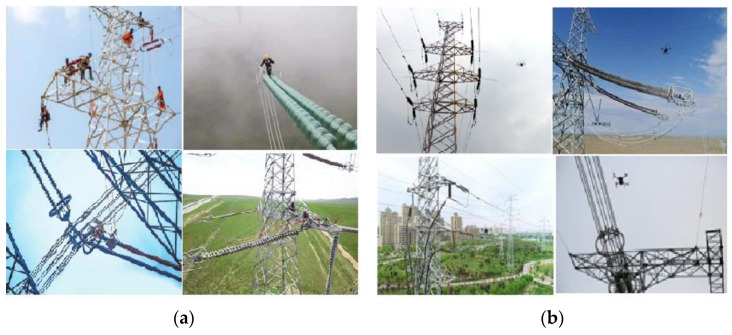
Insulator inspection. (**a**) Manual inspection. (**b**) UAV inspection.

**Figure 2 sensors-23-00272-f002:**
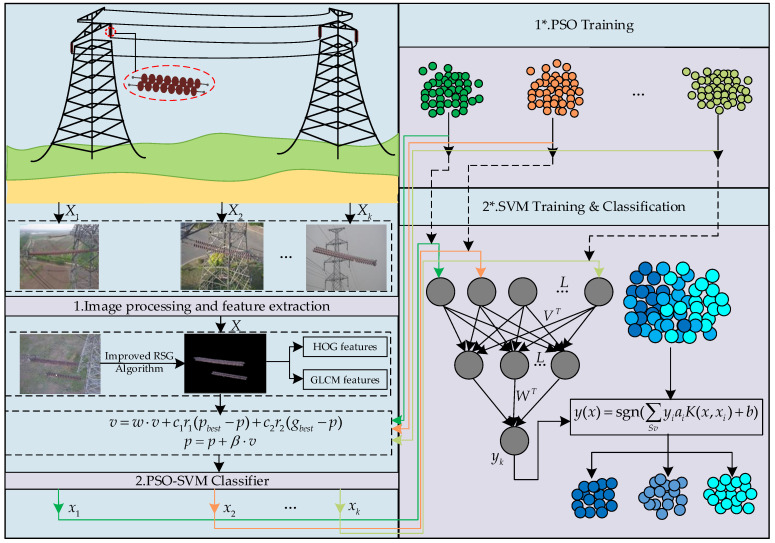
Schematic diagram of the proposed method for insulator state detection.

**Figure 3 sensors-23-00272-f003:**
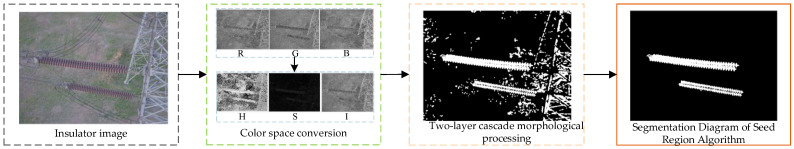
Region segmentation algorithm based on morphological improvements.

**Figure 4 sensors-23-00272-f004:**
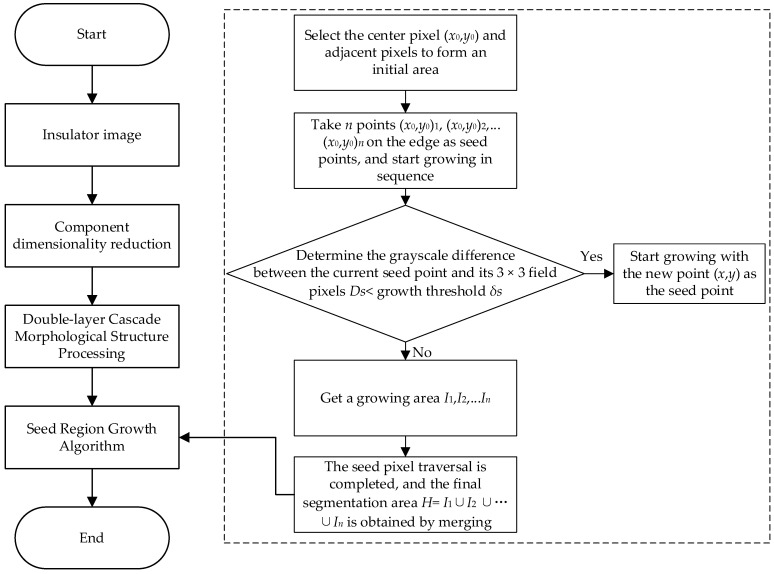
Flow chart of seed region segmentation based on morphological improvement.

**Figure 5 sensors-23-00272-f005:**
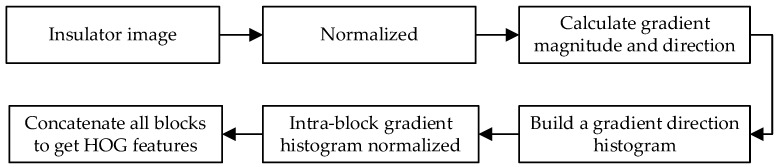
Way to extract the insulator HOG feature.

**Figure 6 sensors-23-00272-f006:**
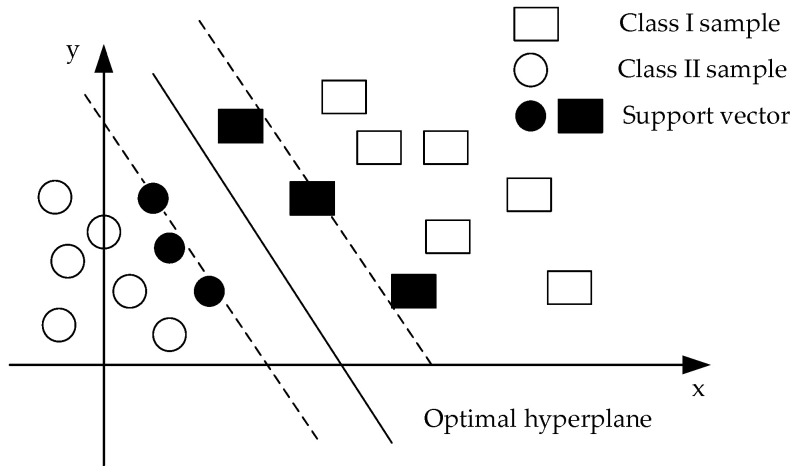
Schematic diagram of the optimal classification surface.

**Figure 8 sensors-23-00272-f008:**
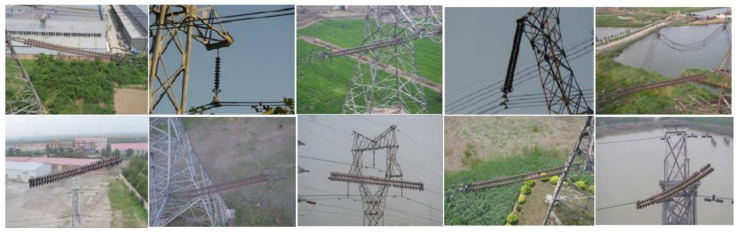
Example of insulator dataset.

**Figure 9 sensors-23-00272-f009:**
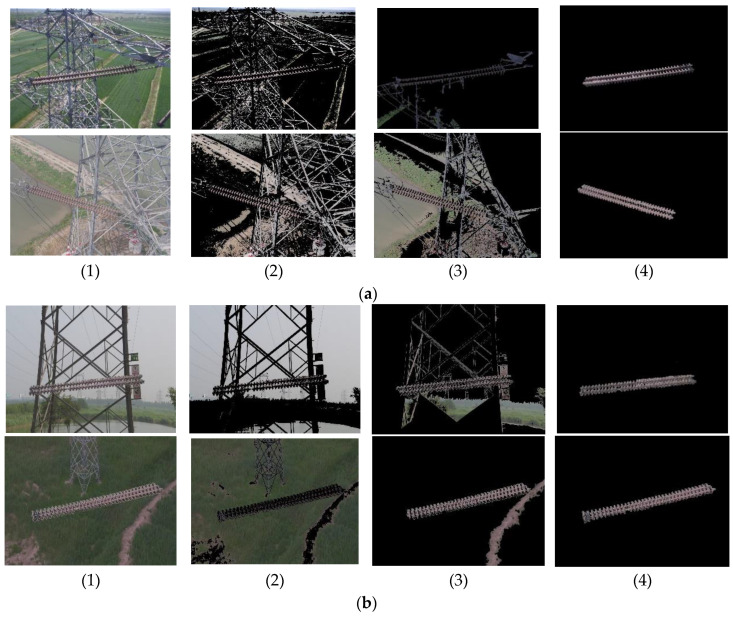
Comparison of image segmentation results of insulator example images. (**a**) Segmentation of normal insulator images. (**b**) Segmentation of defective insulator images.

**Figure 10 sensors-23-00272-f010:**
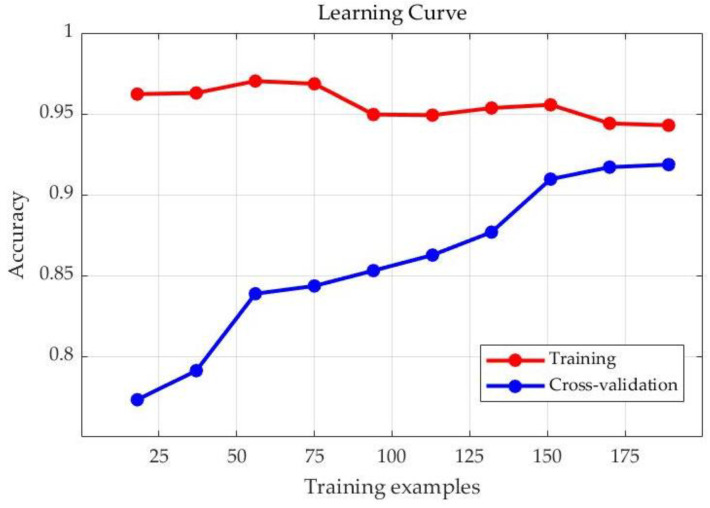
PSO-SVM model learning curve.

**Figure 11 sensors-23-00272-f011:**
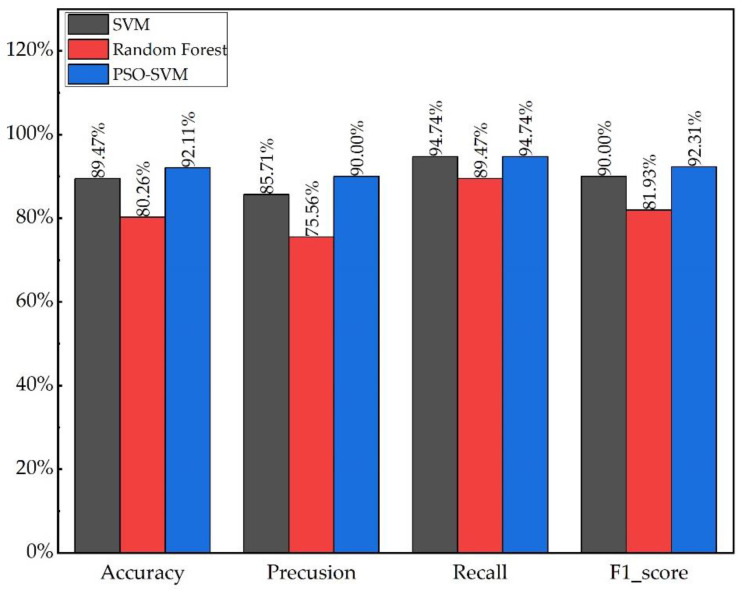
Comparison of classification performance between PSO-SVM and other models.

**Figure 12 sensors-23-00272-f012:**
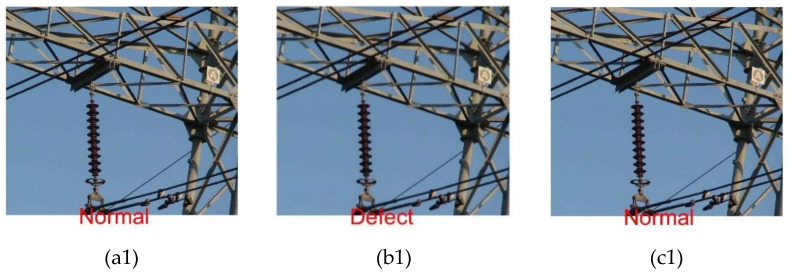
Classification results of each model for image recognition (**a**) SVM; (**b**) Random Forest; (**c**) PSO-SVM.

**Figure 13 sensors-23-00272-f013:**
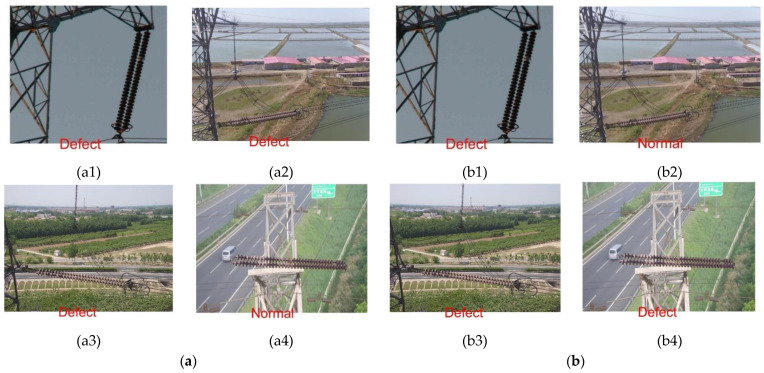
Image recognition classification results: (**a**) GWO-SVM; (**b**) PSO-SVM.

**Figure 14 sensors-23-00272-f014:**
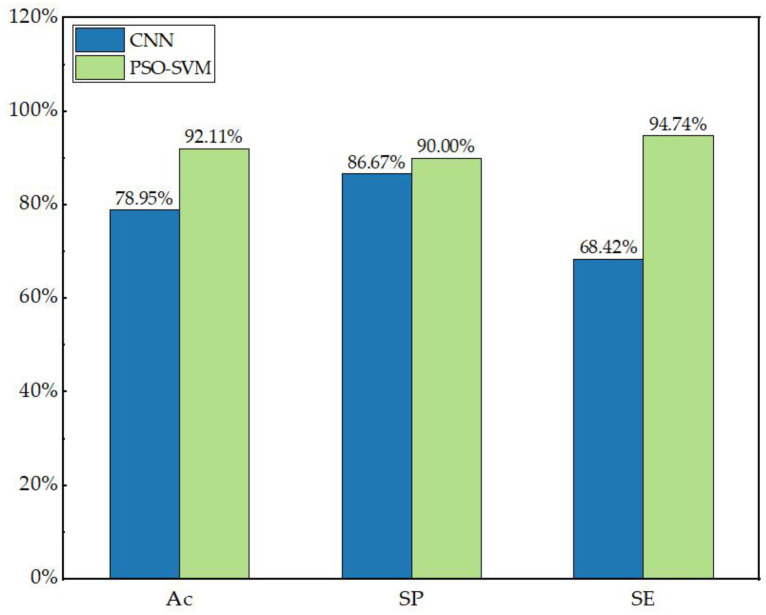
Classification comparison of PSO-SVM and neural network model.

**Figure 15 sensors-23-00272-f015:**
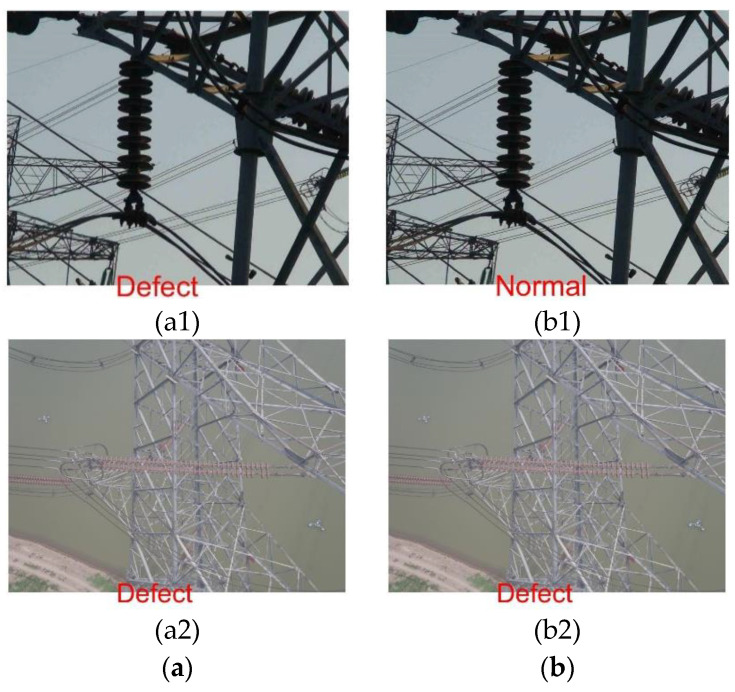
Classification results of each model test set: (**a**) CNN; (**b**) PSO-SVM.

**Table 1 sensors-23-00272-t001:** Literature summary of SVM studies.

Literature (Year)	Author	Research Object	Research Methods	Experimental Results
[39] (2016)	Zhao et al.	Image recognition of infrared insulators with complex backgrounds	Binary feature pools represent insulator shape features, which are identified under SVM.	Accuracy: 89.1026%
[40] (2017)	Zhao et al.	Insulator Pipe Inspection	An advanced discriminative convolutional neural network (CNN) extracted the features of the insulators and trained an SVM for classification.	Accuracy rate: 93%
[41] (2019)	Wang et al.	Classifications of Welding Quality of 100 Resistance Spots	Feature extraction in time domain, frequency domain and wavelet domain were performed on the detected signal, and a particle swarm optimization support vector machine (PSO-SVM) classifier was constructed.	Accuracy rate: 95%
[42] (2020)	Wang et al.	Diagnosis of insulation defects in OIP bushings (defects such as aging, moisture)	The cuckoo search (CS) algorithm optimized the parameters of the multi-class LS-SVM, used the frequency domain dielectric spectroscopy (FDS) method to obtain the data features, and built the multi-class LS-SVM model.	Accuracy rate: 96.25%
[43] (2022)	Cho et al.	Tomato maturity classification in hydroponic greenhouses (green, broken, turned, pink, light red and red)	PCA processed the tomato data and allowed a support vector machine (SVC) to train the model.	Accuracy rate: 75% F1 score: 86%.
[44] (2020)	Zhang et al.	Identification of GIS fault type in insulation fault	The energy entropy features of partial discharge wavelet packets of gas-insulated switchgear (GIS) on high-voltage guide rods are extracted for fault identification under SVM.	Accuracy: 98.125%
[45] (2020)	Van et al.	Bearing Fault Diagnosis	The Max-Relevance and Min-Redundancy (mRMR) method was used to establish feature subsets, and the particle swarm optimization least square wavelet support vector machine (PSO-LSWSVM) classifier was constructed.	Accuracy: 95.33%
[46] (2022)	Chen et al.	Centrifugal Pump Troubleshooting	Continuous wavelet transform (CWT) was performed to obtain data, and parallel factor analysis (PARAFAC) method was employed to extract features for training the IPSO-SVM model.	Correct rate: 100%
[47] (2022)	Li et al.	ECG signal recognition	Unscented Kalman filter denoising was combined with wavelet localization method to detect feature points and compose feature data for training IPSO-SVM model.	Average accuracy: 95.17%
[48] (2022)	Moon et al.	Identification of single target vapors of NO2, HCHO and NH3 and their mixtures	SVM classifier trained with artificial steam database.	Recognition rate: 95.24%
[49] (2022)	Li et al.	Irrigation level classification of sugarcane	Five spectral features highly correlated with irrigation level were used as feature input to construct SVM classifier.	Accuracy rate: 80.6%

**Table 2 sensors-23-00272-t002:** Dataset samples.

Dataset	Training Set	Test Set	Total
Normal image	106	38	144
Defect image	106	38	144
Total	212	76	288

**Table 3 sensors-23-00272-t003:** Experimental environment configuration.

Designation	Version
Operating system	Windows 10 64 bi
CPU	11th Gen Intel(R) Core (TM) i5-1135G7 @ 2.40 GHz 2.42 GHz
Memory	16.0 GB
MATLAB	R2020a
LIBSVM	3.24

**Table 4 sensors-23-00272-t004:** Classification performance of three features in PSO-SVM.

Feature Extraction Algorithm	HOG	GLCM	HOG + GLCM
*C*	3.73	4.77	17.90
σ	0.10	9.80	0.53
Average time	80.28 s	85.35 s	92.58 s
Accuracy	78.95%	50.00%	92.11%

**Table 5 sensors-23-00272-t005:** Classification results of test sets on different models.

Models	Number of Misclassifications/Numbers		Accuracy Rate/%	Precision Rate/%	Recall Rate/%	F1-Score/%
	Normal Sample	Defective Sample				
SVM	2	6	89.47	85.71	94.74	90
Random Forest	4	11	80.26	75.56	89.47	81.93
PSO-SVM	2	4	92.11	90	94.74	92.31

**Table 6 sensors-23-00272-t006:** Comparison of different optimization methods.

Model	Accuracy Rate	Precision Rate	Recall Rate	F1-Score	Average Time
GWO-SVM	81.58%	81.58%	81.58%	81.58%	78.24 s
PSO-SVM	92.11%	90%	94.74%	92.31%	92.58 s

**Table 7 sensors-23-00272-t007:** Insulator image classification results.

	GWO-SVM		PSO-SVM	
	Normal Insulator Diagram	Defective Insulator Diagram	Normal Insulator Diagram	Defective Insulator Diagram
Normal Insulator Diagram	31	7	36	2
Defective insulator diagram	7	31	4	34

**Table 8 sensors-23-00272-t008:** Classification results of the test set on different models.

Model	Accuracy Rate	Sensitivity	Specificity	Average Time
CNN	78.95%	86.67%	68.42%	98.77 s
PSO-SVM	92.11%	90%	94.74%	92.58 s

## Data Availability

The data presented in this study are available on request from the corresponding author. The data are not publicly available due to restriction of privacy.

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
