# Peer review of "A New Approach to Optimize SVM for Insulator State Identification Based on Improved PSO Algorithm"

_sensors, 2022, doi:10.3390/s23010272_

Round 1

Reviewer 1 Report

This paper presents a SVM optimization algorithm for insulator state identification based on improved PSO algorithm, and insulator regions are effectively extracted using an improved region segmentation algorithm based on morphology, and the HOG and GLCM features of the insulator are extracted for fusion. Then, the PSO algorithm is improved and a PSO-SVM classifier is constructed to recognize the insulator state, which improves the detection speed and detection accuracy of recognition to a certain extent. However, there are still some shortcomings:

1. In section 4experiment and result analysis, only the recognition and classification result data of the model is available in each comparative experiment part, and the recognition and classification result image can be added appropriately to help illustrate the effectiveness of the algorithm in this paper.

2. In section 4.4.2 of the paper, the author can give a brief description of the test results in FIG. 10 to facilitate the reader's quick understanding.

3. In section 2.3.3 of the paper, it is mentioned that Penalty factor and kernel parameter are vital parameters that may affect the accuracy of SVM,however, in the following text, the importance of the two parameter values of the penalty factor and the kernel parameter is not shown, and the impact of the value of these two parameters on the accuracy of model detection is not shown. In section 4.3, what is the significance of putting the two parameters in Table 4 or Table 5? It is suggested that the author explain it.

4. In section 4.4 of the paper, Figure 14 shows the classification comparison results of the improved PSO-SVM algorithm and the neural network model proposed in this paper. It is suggested that the author place it in section 4.4.4 instead of section 4.4.3.

5. The insulator data set used in this paper is the public data set on GitHub website. In subsequent research, the author can replace the data set or try to build his own data set to verify the effectiveness of the algorithm proposed in this paper.

Author Response

The reviewer clicks the attachment to review the reply.

Reviewer 2 Report

The paper proposes an approach to optimize an SVM model using an improved PSO algorithm for Insulator State Identification. The idea does not sound original since using metaheuristics to improve machine learning models is not new. Moreover, the following drawbacks must be addressed in a future version:

i) Even though the results seem promising, the experiments are not too convincing because the dataset is too small, so I do not know if the author chose the best images for training and testing the model. Considering Figures 3 and 9, I believe that the best pictures were used.

ii) How the authors show results in Figures 10, 11, 13, 14, and 15 are not common in machine learning. They must show the confusion matrix and metrics using a simple table instead of bar graphs. Thus, we can see the differences clearly. Additionally, figures showing accuracy and loss during the training test help the reader to identify if the algorithm is not overfitting, mainly because the number of samples is small.

iii) How the author chose the PSO configuration in lines 366 to 370? Weirdly, the number of particles is only two. This means that one particle is the best, and the other possibly performs the exploration. Consequently, the loss of diversity will be fast. The idea of having a population is to explore and exploit the search space simultaneously, which is impossible using only two particles. Furthermore, using two particles is something slightly better than Simulated Annealing. In other words, using two particles does not make any sense.

iv) The authors argue that the disadvantage of CNNs models is the need for a huge dataset; however, they compare their approach against Random Forest, which suffers from the same problem because it is a deep learning model.

v) When comparing GWO-SVM vs. PSO-SVM is not clear whether the comparisons were performed under the same conditions. I mean, using, for example, the same number of calls to the objective function.

vi) When comparing the ML models, it is unclear what kind of resampling the authors use. I know that image datasets are computationally expensive, but the dataset is relatively small; thus, the authors should have used a k-fold with at least k=5.

vii) When comparing the PSO-SVM against ANN, some questions remain open. Firstly, what was the architecture used on each ANN? How was the CNN optimized? Did they use Adam, AdaGrad, AdaDElta, or RMSProp? Why did the authors not use known architectures such as Yolo, ResNet, or others? By the way, BP is not a kind of ANN is a training algorithm.

Considering all those drawbacks, I have to consider the article rejected.

Author Response

(The authors gave the same response as above.)

Round 2

Reviewer 1 Report

The author has carefully revised and improved the article according to the review comments, and the quality of the manuscript has been greatly improved, and it is recommended to be accepted.

Author Response

Please see the attachment for our reply to Reviewer 1

Reviewer 2 Report

The authors improved the paper quality significantly. However, Section 4.4.4 is still a mystery. Firstly, they do not describe the CNN or ANN architectures. Also, they did not describe the hyperparameters. Consequently, the experiment is not reproducible. Secondly, again they are calling BP an ANN, which is incorrect. It is essential to address this methodology issue before publication. 

Author Response

We reply to each reply of reviewer 2 one by one, please see the attachment.
